# A Multilevel Analysis of the Associated and Determining Factors of TB among Adults in South Africa: Results from National Income Dynamics Surveys 2008 to 2017

**DOI:** 10.3390/ijerph191710611

**Published:** 2022-08-25

**Authors:** Hilda Dhlakama, Siaka Lougue, Henry Godwell Mwambi, Ropo Ebenezer Ogunsakin

**Affiliations:** 1Department of Statistics, University of Johannesburg, Johannesburg 2028, South Africa; 2School of Mathematics, Statistics and Computer Sciences, University of KwaZulu-Natal, Durban 4041, South Africa; 3Discipline of Public Health Medicine, School of Nursing and Public Health, University of KwaZulu-Natal, Durban 4041, South Africa

**Keywords:** TB, multilevel models, generalized linear mixed-effects models, South Africa, National Income dynamics survey, Bayesian analysis

## Abstract

TB is preventable and treatable but remains the leading cause of death in South Africa. The deaths due to TB have declined, but in 2017, around 322,000 new cases were reported in the country. The need to eradicate the disease through research is increasing. This study used population-based National Income Dynamics Survey data (Wave 1 to Wave 5) from 2008 to 2017. By determining the simultaneous multilevel and individual-level predictors of TB, this research examined the factors associated with TB-diagnosed individuals and to what extent the factors vary across such individuals belonging to the same province in South Africa for the five waves. Multilevel logistic regression models were fitted using frequentist and Bayesian techniques, and the results were presented as odds ratios with statistical significance set at *p* < 0.05. The results obtained from the two approaches were compared and discussed. Findings reveal that the TB factors that prevailed consistently from wave 1 to wave 5 were marital status, age, gender, education, smoking, suffering from other diseases, and consultation with a health practitioner. Also, over the years, the single males aged 30–44 years suffering from other diseases with no education were highly associated with TB between 2008 and 2017. The methodological findings were that the frequentist and Bayesian models resulted in the same TB factors. Both models showed that some form of variation in TB infections is due to the different provinces these individuals belonged. Variation in TB patients within the same province over the waves was minimal. We conclude that demographic and behavioural factors also drive TB infections in South Africa. This research supports the existing findings that controlling the social determinants of health will help eradicate TB.

## 1. Introduction

TB and the human immunodeficiency virus (HIV) form a deadly synergy. In 2017, the WHO reported that 78,000 people died, of which 56,000 were HIV positive [1]. In addition to early diagnosis and effective treatment of TB, the effective use of Anti-retroviral Therapy (ART) for HIV has contributed to the decline in TB deaths Globally. The rank of TB (TB) as the leading cause of death is declining over time, but in South Africa, TB has maintained its position as the leading cause of death. The South African National TB Programme (NTP) has put interventions in place intending to “bring down the national or global incidence from more than 1000 per million population in 2015 to less than 100 per million by 2035” [2,3].

Several references, some of which we quoted in this study, state that TB is a disease of poverty. Ref. [4] demonstrates this link using a potential causal pathway for low income and TB In their study, they used the following variables at two levels, Individual: Age, sex, education, race, smoker, alcoholism, Body Mass Index (BMI), employment, and at the household level: Urban residence, number of adults per bedroom, affordability of meals and household asset score. In their study on TB in Western Cape, South Africa, [5] identified the risk factors for infection: overcrowding, number of infectious cases in the community, poor nutritional status, alcoholism, and unemployment. Medication adherence was also a contributing factor, so they recommended that non-medical interventions are vital to the success of TB control programs. Ref. [6] on identifying risk factors of TB/HIV in South Africa, 2006 reported those married or have surviving partners to be at lower risk of TB/HIV. A National study [7] said that Eastern Cape Province is the hardest hit by TB and Limpopo province the least. They also listed poor living conditions, lower socio-economic status, English illiteracy, lack of Secondary/Tertiary education, alcohol consumption, marital status, age groups, and gender as drivers of TB.

The inconsistency in TB deaths is ascribed to socio-economic factors associated with place of birth, income, education and healthcare access, and regional differences [8,9] also concluded that it is because of their overexposure to poor living conditions in overcrowded places with inadequate hygiene, protection, and malnutrition. TB and general health status also depend more on individual risk factors such as age, sex, smoking, alcoholism, diabetes, HIV status, marital status, ethnicity, homelessness, drug use, and migrant status. Other socio-economic and environmental risk factors include deprivation, financial insecurity, and housing conditions. In their multilevel cross-sectional data analysis on self-reported TB for a sample in Eastern Cape, South Africa [10], also recommended the need to consider possible benefits of programs that deal with housing and social environments when addressing the spread of TB in economically poor districts.

This research was motivated by the burden of TB in South Africa. Although most disease modeling in Statistics in medicine is based on clinical trials, observational studies have also helped identify new interventions to curb TB A few survey studies have explored the multilevel TB models in South Africa, but very few included the repeated measures component. This research is also in line with the Stop TB ‘partnership’s “Zero TB initiative”, whose purpose is to create “islands of elimination” by identifying communities at risk and recommending models of intervention [2]. In this regard, ref [11] used transmission models to determine if these targets were reachable in a bid to eradicate TB in South Africa. They also concur with refs [7,12] that eradicating this disease will only occur if other interventions apart from treatment are implemented. Therefore, it is vital to include social, economic, and environmental determinants of TB in the strategy to stop TB deaths and new infections.

## 2. Materials and Methods

### 2.1. The NIDS Data

The NIDS data is a nationally representative sample survey of 28,000 individuals in 7300 households across South Africa that has been tracked since 2008. The NIDS data is accessible in the public domain (http://www.nids.uct.ac.za/nids-data/data-access (accessed on 16 September 2019)). To date, the NIDS data has five waves, Wave 1 (2008), Wave 2 (2010), Wave 3 (2012), Wave 4 (2014/2015) and Wave 5 (2017). A detailed description of this data is given elsewhere [13]. The sample designs used for the NIDS data organize populations into clusters of “natural units” [14], for example, provinces and households, before collecting data within the clusters, and this leads to correlated data [15]. This structure gives the data a multilevel, also called hierarchical nature, increasing standard errors due to homogeneity. The clusters give rise to within and between clusters variation. Apart from growing standard errors due to uniformity, clusters also give rise to within and between clusters variation. Biased parameters also arise if such data is analyzed using models that do not consider the clustered nature [16]. “The existence of such hierarchies is neither accidental nor ignorable”, and such multilevel data cannot be treated as standard cross-sectional or longitudinal data [17].

The hierarchical structure of the NIDS data has repeated observations (waves) nested within individuals, which are nested within households, geographical types (rural or urban), district councils, and provinces. Since the data is hierarchical due to sampling, Multilevel Models (MLM) will account for the correlations [11,18]. Multilevel risk factors of TB were analysed at different levels, which will assist in studying the variations among the other individuals and provinces and their influence on self-reported TB infections. The MLMs, as proposed by [18], consider the clustering and stratification effects [19]. Multilevel refers to a hierarchical or nested structure. “MLM or Multilevel analysis is used as a generic term for all models for nested data” [20]. It can be individuals nested within groups or repeated measures nested within individuals focusing on change/growth or relationships between variables within an individual. In hierarchical data, missing data is bound to arise as dropouts (measurement sequences terminated) or intermittent missing data (missing intermediate scheduled measurements). Still, one of the advantages of MLM approaches is the ability to handle unbalanced data by incorporating random effects of varying degrees of complexity depending on the multilevel structure.

### 2.2. Model Formulation

A form of MLMs is the Generalised Linear Mixed Model (GLMM). GLMM is an extension of Generalised Linear Model (GLM) that models a mean response depending on the fixed effects and also on the random effects. For a fixed-effects factor, we assume a finite set of group levels that are of interest and the inferences are to be made only concerning those levels. For a random-effects factor, however, we assume that there is an infinite set of levels present in the study from that population and only a random sample of them are included in the study. As fixed effects are interpreted as regression coefficients, random effects can be interpreted as random intercepts or random slopes and the extra variability captured via the corresponding variance components which are to be estimated from the marginalised model.

For a two-level hierarchy, where there are $\overset{n}{\underset{i=1}{\sum }}m_{i}\;$ subjects over the *n* clusters where *m_i_* is the number of subjects in the *i*th cluster with $Y_{ij}$$\left(j=1,\ldots ,m_{i};i=1,\ldots ,n\right)$ representing the *j*th outcome for cluster *i*. The outcomes $Y_{ij}$ are conditionally independent with conditional densities of the form of Equation (1) given by [21,22]:(1)$$f(y)=f(y_{ij}|\gamma_{i},\vartheta ,\varphi )=\exp\left\{\frac{\left(y_{ij}\vartheta_{ij}-b(\vartheta \right)}{\varphi }+c(y,\varphi )\right\}$$
(2)$$\mathrm{With}\;\eta (\mu_{ij})=\eta [E(Y_{ij}|\gamma_{i})]=x_{ij}^{T}\beta +z_{ij}^{T}\gamma_{i}$$
for a known link function η(.) where $x_{ij}$ and $z_{ij}$ are p-dimensional and q-dimensional vectors of known covariate values, with β a p-dimensional vector of unknown fixed regression coefficients, with a scalar parameter φ. The density of the $N(0,\sum_{\gamma })$ distribution for the random effects $\gamma_{i}$ is $f(\gamma_{i}|\sum )$.

The most popular GLMM for analysis of clustered binary coded data is the generalised mixed-effects logistic regression model. This model uses the same model as (4) where $X_{i}$ represents level-1 fixed effects covariates and $Z_{i}$ represents level-2 or more random effects and their coefficients for all the subjects. In general, we introduce a link function $g(\mu_{ij})=\eta_{i}$, which links the expected response $\mu_{ij}$ to the linear predictor, where
(3)$$\eta_{i}=X_{i}\beta +Z_{i}\gamma_{i}$$

For binary coded data, we use a logit link function $g(\mu_{ij})=\log\left(\frac{\pi_{ij}}{1-\pi_{ij}}\right)$.

Equation (2) can then be expressed in matrix notation as
(4)$$g\left(\mu_{ij}\right)=[E(y_{ij}|\gamma_{i})]=X_{ij}\beta +Z_{ij}\gamma_{i}$$
where:

$Y_{i}$ is a $n_{i}\times 1$ column vector of responses for $n_{i}$ available observations,$X_{i}={\left(X_{i1,\ldots ,\;}X_{in_{i}}\right)}^{T}$ the desgn matrix for the fixed effect with $X_{ij}={\left(1,X_{ij1,\ldots ,\;}X_{ijk}\right)}^{T}$,β is a p×1 column vector of unknown regression coefficients for the fixed effects,$Z_{i}={\left(Z_{i1},\ldots .,\;Z_{in_{i}\;}\right)}^{T}$ and design matrix for the random effects with $Z_{ij}={\left(1,Z_{ij1},\ldots .,\;Z_{ijq\;}\right)}^{T}$,$\gamma_{i}$ is a q×1 vector of unknown subject-specific random effects representing all levels of single factor where, γ$\sim N(0,\sum_{\gamma })$ and $\sum_{\gamma }$ is a q×q positive-definite matrix.

Here $\mu_{ij}=E(Y_{i}|\gamma_{i})$ is the conditional expectation of $Y_{i}$ given $\gamma_{i}$ and is equal to $P(Y_{i}|\gamma_{i})$, the conditional probability of a response given the random effects (and covariate values) and is generally a member of an exponential family.

The variance of Y, $Var(Y)=Z^{T}Var(\gamma )Z+Var(\in )=Z^{T}\Sigma_{\gamma }Z+\Sigma_{\in }$.

To determine how much of the total variance between groups, the Intraclass Correlation Coefficients (ICC) must be obtained at every level of the random intercept models.

We consider one case of MLM where data is modelled cross-sectionally by waves, with two levels, the individual level as level 1 nested into level-2 provinces. We, therefore, have a random intercept for provinces. The main focus in this context is on the individual and the variation caused by belonging to different provinces.

For simple formulation, we used random intercept models with no interactions.

We let $Y_{ij}$ denotes the dichotomous response variable, taking value 1 for a positive TB response for a person *i*, i=1,2,…,n nested within the *j*th province j=1,2,…,9. Assuming a value of 0 in the absence of TB. β is a vector of unknown regression coefficients for the fixed effects. γ is a vector of random effects for level-2.

This study focused on self-reported TB from 2008 to 2017 in South Africa: adults (above age 15). The outcome variable of interest is those who self-reported to have been TB diagnosed. That is individuals who were diagnosed with TB or not. The independent variables used were as follows: marital status: married, single (β_1_), age: 15–29, 30–44 (β_2_), 45–59 (β_3_), 60+ (β_4_), gender: male, female (β_5_), race: African, non black-African (β_6_), education: none, primary (β_7_), secondary (β_8_), Tertiary (β_9_), home language literacy: can read home language, cannot read (β_10_), employment status: employed, not employed (β_11_), regular smoking: yes, no (β_12_), Suffer from other diseases: yes, no (β_13_), regular exercise: yes, no (β_14_), consult with health practitioner in the last two years: yes, no (β_15_), Ever been diagnosed with Asthma: yes, no (β_16_), Ever been diagnosed with diabetes: yes, no (β_17_), household received any form of social grant: yes, no (β_18_), perceived household income above average: yes, no (β_19_), access to better housing: yes, no (β_20_), household income below median: yes, no (β_21_), household expenditure below median: yes, no (β_22_), geographical type: traditional, urban (β_23_), or farms (β_24_).

For the variable, “Suffering from other diseases” meant other diseases such as Physically handicapped, Problems with sight, hearing or speech, Psychological or psychiatric disorder, HIV/AIDS, Epilepsy/fits, Emphysema and Alzheimer’s disease. For the variable “Access to better housing”, we created a proxy variable using the Principle Component Analysis (PCA) method. We used the type of material used to build a dwelling (floors, walls and roof) and the type of energy used (for cooking and lighting). We took the first component and used it to make two quintiles, where the upper quintile was coded as 1 for households with access to better housing and the lower quintile as households with no access to better housing.

We did a stepwise regression to determine the most influential risk factors to determine the validity of using MLM, as outlined by [15]. We started with the logistic regression model for the fixed effects factors selected by stepwise regression, followed by a two-level random intercept model with the same fixed effects and province as a random effect. For MLMs, like any regression model, it is essential to check if the assumptions associated with that model are satisfied. Since our MLM is logistic, we cannot test for normality, but there is a need to check if there is no perfect linear relation among the predictor variables. If predictor variables are linearly correlated, the estimates for that model cannot be uniquely computed. When more than two variables are included in a model, and there is a linear correlation among them, it is called multicollinearity. As multicollinearity increases, the model coefficients become unstable and standard errors inflated. Since this study is based on two levels of analyses, province and individual-level, we had to check for multicollinearity among the predictor variable. We performed a Variance Inflation Factor (VIF) analysis in STATA 15. VIF denotes the percentage of variance in one predictor described by other predictors in the model. As a rule of thumb, a variable whose VIF values are more significant than ten may merit further investigation [23,24]. Tolerance, defined as 1/VIF, is used by many researchers to check the degree of collinearity. A tolerance value lower than 0.1 is comparable to a VIF of 10. It means that the variable could be considered as a linear combination of other independent variables.

The frequentist and Bayesian approaches were both used for inference. Non-informative priors were used to model Wave 1 to 4 data. Elicitation of prior knowledge from historical data was done to obtain informative priors to use for Wave 5. The elicitation was done by averaging the individual posterior density parameters obtained for Wave 1 to 4 using priors the following priors, for Beta, mean 1 and variance 0.0001, for Beta.province, mean 0 and variance tau.prov where tau.prov had a mean of 0.01 and variance 0.01. The results for frequentist and Bayesian approaches were compared. The primary purpose of logistic regression is to give an overview of the relationships and their strengths.

## 3. Results

The samples used for the five waves were of size 17,102, 18,585, 21,313, 23,487, and 25,076, respectively, with the proportion of people who reported being TB diagnosed for these samples being 4.5%, 3.6%, 5.3%, 5.7%, and 5.1% for wave 1 to wave 5 respectively.

### 3.1. Multicollinearity Results

As seen in Table 1, All VIFs are below 10, which means there was no collinearity among the predictor variables. The average VIF was 1.85.

### 3.2. Frequentist MLM Results

This section emphasizes the variables that differ in significance and the province’s impact as a population level on the determinants of TB that we identified from the descriptive statistical analysis and the stepwise regression in STATA 15. These are shown in Table A1, in Appendix A. The MLMs were valid and fit the data better than logistic models for all five waves. The variances show this contributed at the province level, the Likelihood Ratio Test (LRT), Wald test, BIC(Bayesian Information Criterion), and AIC(Akaike Information Criterion). The ICCs indicate that level two contributed 4.8%, 3.2%, 4.3%, 5.2%, and 6.2% of the variation in TB infections, respectively, for waves 1 to 5.

The variables that best described the TB situation in 2008 were: marital status, age, gender, education, smoking, suffering from other diseases, exercising, consultation about health, diagnosis with asthma, and perceived income. With an odds ratio of $e^{0.25}=1.3\;$ (coef. 0.25, 95% CI (0.068–0.438)), the single were 1.3 times more likely to have been diagnosed with TB than married people. The odds ratio for age groups 30–44, 45–59, and 60+ were 3.0, 2.5, and 1.5, respectively. This means that compared to the baseline age group of 15–29, these age groups were 3.0, 2.5, and 1.5 times more likely to be diagnosed with TB Females were less likely to be diagnosed with TB than males by 41%. (OR 0.59, coef. −0.52 (−0.719:−0.329)). People with Secondary school as their highest education qualifications were less likely to be diagnosed with TB, coef. −0.41 95% CI (−0.744; −0.074). This means they were approximately 34% (OR 0.657) less likely to be diagnosed with TB than those without education. The non-regular smokers were 40% less likely to be diagnosed with TB than regular smokers, with an odds ratio of $e^{-0.5}=0.6$, coef. −0.5, 95% CI (−0.70; −0.287). People who did not suffer from other diseases were 25% less likely to be diagnosed with TB (coef. −0.28, 95% CI (−0.521; −0.047), OR 0.75). The odds of individuals developing active TB were positively associated with consulting with a health practitioner in the past two years. Those who had not consulted with a health practitioner were 66% less likely to be diagnosed with TB, with an odds ratio of $e^{-1.09}=0.34$ 95% CI (−1.357; −0.819). Those who were diagnosed with asthma had coef. 0.45, 95% CI (−0.788; −0.115) were $e^{0.451}=1.57$ times more likely to be diagnosed with TB than those without asthma. Individuals who belong to households with perceived income below average were 43% (OR: $e^{0.36}=1.43$, 95% CI (0.165; 0.546) more likely to be diagnosed with TB.

The factors that showed significant differences in TB infections in 2010 were age, race, smoking, other diseases, consulting with a health practitioner, social grant, and geographical type. Those aged 30–44 had an odds ratio of $e^{1.185}=3.277$ (coef. 1.185, 95% CI (0.916; 1.454) means they were 3.3 times more likely to contract TB than those aged 15–29. Those aged 45–59 were 2.6 times (OR 2.578, 95% CI (0.627; 1.267)) more likely to contract TB than the 15–29-year-olds. The 60+ had a lesser chance of contracting TB than the 30–59-year-olds but were 1.6 times more likely to contract the disease than the 15–29 years old. Non-black South Africans were 36% less likely to be diagnosed with TB than black South Africans, OR: $e^{-0.445}=0.641$, 95% CI (−0.783; −0.107). Non-regular smokers were 48% less likely to be diagnosed with TB than non-smokers OR:$\;e^{-0.659}=0.517\;$ (95% CI (−0.917; −0.400)). People who did not suffer from other diseases were 43% less likely to be diagnosed with TB than those who suffered from other diseases, OR: $e^{-0.571}=0.565$ (coef. −0.571, 95% CI −0.933; −0.210). With an OR of 0.336 (coef. −0.09, 95% CI −1.358;−0.822), people who had not consulted with a health practitioner had a 67% lesser chance of being diagnosed with TB than those who had not consulted. The OR for those who were not getting some form of a social grant was $e^{-0.282}=0.754$ (coef. −0.282, 95% CI −0.519; −0.045) means people who belonged to households that did not receive some form of a social grant had a 25% lesser chance of contracting TB compared to those who did receive a social grant. People residing in urban areas were 1.6 times more likely to be diagnosed with TB than those in traditional dwellings with OR: $e^{0.482}=1.62$ (coef. 0.482, 95% CI 0.225; 0.738).

In 2012, factors associated with TB were: marital status, age, gender, education, employment status, smoking, other diseases, consult with a health practitioner, asthma diagnosis, access to better housing, and geographical type. Single people were 1.3 times more likely to contract TB than their married counterparts. With OR of 3.277, 3.194, and 1.995 for those aged 30–44, 45–59, and 60+ respectively, the odds of contracting TB were 3.277, 3.194, and 1.995 times as large as the odds for individuals in the age group 15–29. The odds for TB were less for individuals who have been to a tertiary school than those with no education or just primary education (OR 0.439). With an OR of 1.198, the unemployed were almost 1.2 times more prone to TB than the employed. An OR of 0.609 means non-smokers had a 40% less chance of contracting TB than their smoking counterparts with coef. −0.495, 95% CI (−0.670; −0.321). people who suffered from other diseases had an OR of $\;e^{1.064}=$ 2.90, meaning the odds of contracting TB were 2.9 times more than those who did not suffer from any other disease. Those who had not consulted with a health practitioner in the last two years had lesser odds of contracting the disease, OR 0.345, coef. 95% CI (−0.808; −0.448). There was a positive association between TB and having asthma, OR 0.442 for not having asthma means people who had never been diagnosed with asthma had 56% less chance of being diagnosed with TB than those who had been diagnosed with TB. On access to better housing, individuals that belonged to households with no access to better housing had lower odds of contracting TB than those with access to better housing, 95% CI (−0.372; −0.059), OR 0.806. the odds of contracting TB for people residing in urban areas were 1.3 times more than those in traditional dwellings.

In wave 4 (2014/2015), the factors associated with TB were marital status, age, gender, education, employment status, smoking, other diseases, consultation with a health practitioner, asthma, perceived income, household income, and geographical type. Single people were approximately 1.4 (coef. 0.359, 95% CI (0.221; 0.497)) times more likely to have been diagnosed with TB than their married counterparts. The odds of being diagnosed with TB for age groups 30–44 and 45–59 were approximately 3.2 times higher than those aged 15–29. For the senior citizens aged sixty and above, the OR was 1.705 ($e^{0.534}=1.705\;$ (coef. 0.534, 95% CI 0.295; 0.773)). Females were 24% less likely to be diagnosed with TB than males (coef. −0.269, 95% CI −0.407; −0.132). In education, only tertiary education was significantly different from no education as far as TB was concerned. With an OR of 0.65, a person with tertiary education as their highest qualification was 35% less likely to be diagnosed with TB than one without education. With coef. −0.438, 95% CI (−0.805; −0.070). The non-employed were 1.2 times more likely to be diagnosed with TB than the employed (coef. 0.154, 95% CI (0.009; 0.298)). Non-regular smokers had a 29% (coef. −0.337, 95% CI (−0.492; −0.182)) have less chance of being diagnosed with TB than regular smokers. There was an association between TB and other diseases, consultation, and asthma. People who do not suffer from other diseases had not consulted with a health practitioner and had not been diagnosed with asthma were 59%, 62%, and 48% less likely to be diagnosed with TB, respectively (odds ratios 0.41, 0.38, and 0.52 respectively). There was a positive relationship between perceived income, household income, and geographical type. People who perceived their household income to be below average were 1.2 times more likely to be diagnosed with TB compared to those who perceived their household income to be above average with OR: $e^{0.161}=1.175$ (95% CI (0.036; 0.287)). Individuals belonging to households with income above the median had higher odds of being diagnosed with TB than people whose household income was below average or median. The odds ratio was approximately 1.2 (95% CI (0.035; 0.338)).

In 2017 TB was positively associated with marital status, age, gender, race, regular smoking, other diseases, regular exercises, consult with a health practitioner, asthma, perceived household income, and household expenditure. The singles were 1.3 times more prone to TB infection than their married counterparts OR: $e^{0.276}=1.318$ (coef. 0.276, 95% CI (0.133; 0.420)). With an OR of 3.494, those aged 30–44 years were 3.5 times more likely to contract TB than those 15–29 years old. Slightly higher odds for those aged 45 to 59; they were 3.6 times more likely to develop TB than those aged 15–29 years ($e^{1.293}=3.644$, 95% CI: (1.082; 1.504)). With an OR of 2.2, those over 60 years were twice as likely to contract TB as the 15–29-year-olds. Non-black Africans were 23% less likely to contract TB compared to their black African counterparts, with an OR: $e^{-0.26}=0.771$, 95% CI (−0.475; −0.044). Non-regular smokers were 26% less likely to be diagnosed with TB compared to regular smokers (OR: $e^{-0.308}=0.735$, coef. −0.308, 95% CI (−0.469; −0.147)). People who did not suffer from other diseases had a 65% lesser chance of TB diagnosis than those who suffered from other diseases with an OR of 0.35, 95% CI (−1.216; −0.886). An OR of 1.2 for regular exercise meant that those that did not exercise regularly were 1.2 times more likely to be diagnosed with TB than those who exercised regularly (coef. 0.185, 95% CI (0.031; 0.338)). The odds of getting TB diagnosed for those that had not consulted with a health practitioner was 0.525 (coef. −0.644, 95% CI (−0.843: −0.445)) and 0.531 for those that had not been diagnosed with asthma, meaning they had a 47% less chance of being diagnosed with TB compared to their counterparts. Those who perceived their income to be below average were 1.16 times more likely to be diagnosed with TB (OR: $e^{0.153}=1.165$, 95% CI (0.019; 0.286)). Those with no access to better housing were 20% less likely to get TB infected ($e^{-0.222}=0.881$, coef. −0.221, 95% CI (−0.374; −0.070)). Not belonging to households with monthly expenditure below the median rendered individuals 1.31 times more likely to be diagnosed with TB than their counterparts.

### 3.3. Bayesian MLM Results

In this section, we present results for Bayesian MLM for all waves using non-informative priors. A comparison of the frequentist MLM and Bayesian with both non-informative and informative priors is also shown in Table A2, Appendix A. In wave 1, the TB-related factors were marital status, age, gender, race, education (secondary education), smoking, suffering from other diseases, exercise, consult with a health practitioner, asthma diagnosis, and perceived household income. In wave 2, age, race, smoking, suffering from other diseases, consulting with a health practitioner, social grant, and geographical type were the determinants of TB In wave 3, the predictors of TB were marital status, age, gender, race, education (tertiary), employment status, smoking, suffering from other diseases, consult with a health practitioner, diagnosis with asthma, and geographical type (urban). The predictors of TB in wave 4 were marital status, age, gender, education (tertiary), employment status, smoking, suffering from other diseases, consult with a health practitioner, asthma, perceived household income, monthly household income, and geographical type (urban). In wave 5, the factors influencing TB were marital status, age, gender, race, smoking, other diseases, asthma diagnosis, perceived household income, access to better housing, and household expenditure. All five models’ variances were significant, meaning the MLMs were valid and more informative than the regression models. Using the posterior means and standard deviations for waves 1 to 4, wave 5 priors were elicited and are shown in Table A3, Appendix A.

### 3.4. Comparison of Posterior Distributions for Informative and Non-Informative Priors

In this subsection, we first compare the two Bayesian models with informative and non-informative priors by using the estimates, MC errors, and DIC values. The informative priors obtained from the posterior distributions of the four waves are shown in Table 2.

Both models indicated the same predictors of TB These were as mentioned in Table A3. The MC errors for the model with non-informative priors are lower than the MC errors for informative priors. The same trend is seen in DICs. Although the difference is slight, the model with non-informative priors had a lower DIC value than the model with informative priors.

We conclude that the informative priors did not improve the Bayesian MLM model for this data. Therefore, in the next section, we will use the Bayesian MLM models with non-informative priors to compare with the frequentist MLM models. This model converged at 100,000 iterations, the density plots were smooth, the BGR diagnostics were approaching 1, and the plots are shown in Appendix B, Figure A1, Figure A2 and Figure A3.

### 3.5. Comparison of Frequentist and Bayesian MLM

The results of the comparison of the two estimation approaches are shown in Table 3. The frequentist and Bayesian approaches identified the same factors affecting TB (Table 4). This disease was positively associated with marital status, race, age, gender, regular smoking, other diseases, regular exercise, asthma, household perceived income, access to better housing, and household expenditure.

## 4. Discussion

In identifying the risk factors of TB diagnosis from 2008 to 2017, this article aimed to determine the factors associated with these self-reported TB-diagnosed patients. As expected, most of the variation in TB exists at the person level (level-1). This paper presented the Multilevel modelling of TB from 2008 to 2017. We considered the individual level grouped within provinces. So Provinces were considered at level-2 as the population grouping level, whilst the individuals were at level-1. We performed two-level models on all waves, 1 to 5, using the two estimation approaches: frequentist and Bayesian. In addition to the individual wave results, for each estimation approach, a summary of TB factors was presented for all the waves, based on statistical significance. The aim was not only to identify TB factors at the individual level but also to determine to what extent the factors vary across such individuals belonging to the same province. To check the performance of the two estimation methods for this kind of data, the two methods were compared to identify the more robust approach. The methodological findings were that the frequentist and Bayesian models resulted in the same TB factors. Both models showed that some form of variation in TB infections is due to the different provinces these individuals belonged. Of importance to note is that the informative priors did not make the Bayesian model a better model, so we conclude based on the Bayesian model with non-informative priors. The Bayesian model showed near-zero MC errors and a slight reduction in Credible intervals compared to the frequentist model’s confidence interval. Our results differ from the findings of [25], who did a similar research but concluded that the Bayesian approach helps select more significant factors related to TB as opposed to the classical approach.

TB factors that prevailed consistently from wave 1 to wave 5 were age, smoking, suffering from other diseases, and consulting with a health practitioner in the past two years. Our overall findings revealed that the significant variables associated with TB were marital status, age, gender, race, unemployment, suffering from other diseases, regular exercise, regular smoking consult about health, diagnosis with asthma, diabetes, housing, household income, and geotype. This research conforms with previous findings that report unemployment [5] and poor living conditions [7,8,9] as the risk factors for TB. We did not take households as a level, but households with income above the median were more at risk of being diagnosed with TB. This result does not conform with most research but conforms with the findings of [26], who did a spatial analysis of TB and found TB to be associated with urbanicity. This could also be because South Africa’s median income is too low and renders households poor. Therefore, although some households have income and expenditure above median, they could still be living under poor economic conditions where TB is rife. Smoking is a known risk factor for TB infection [7,27].

## 5. Conclusions

Some of the key discoveries for this research is that the Bayesian and frequentist estimation approaches yielded the similar results. Using informative priors did not make any difference to the model estimation. Though significant, the variability across individuals belonging to the same province over time was minimal, indicating a weak correlation between individuals in the same province. Finally, the current study proved that, to a significant extent, individual-level characteristics are associated with TB in South Africa. Therefore, policy-makers seeking to reduce provincial inequality in the TB disease should endeavor to identify and address the critical factors at the individual levels of the social environment.

## Figures and Tables

**Table 1 ijerph-19-10611-t001:** Results obtained using Multicollinearity analysis.

Variable	VIF	1/VIF
Marital status: Single	1.3	0.769004
Age: 30–44	1.48	0.6761
45–59	1.7	0.588023
60+	1.74	0.576259
Gender: female	1.27	0.788557
Race: non-black-African	1.38	0.725238
Education: Primary	4.73	0.211383
Secondary	7.46	0.134047
Tertiary	3.98	0.251132
Home language literacy: cannot read	1.98	0.504168
Employment status: not employed	1.35	0.740186
Regular smoking: No	1.25	0.79856
Other disease: No	1.04	0.963998
Regular exercise; No	1.16	0.865169
Consult with health practitioner: No	1.07	0.934643
Diagnosed asthma: No	1.01	0.985332
Diagnosed diabetes: No	1.08	0.922889
Social grant: No	1.18	0.844367
Perceived household Income: No	1.14	0.879005
Access to better housing: No	1.3	0.769982
Income below median: No	1.49	0.671186
Expenditure below median: No	1.55	0.647064
Geotype: Urban	1.46	0.684569
Farms	1.2	0.836529
Mean VIF	1.85	

**Table 2 ijerph-19-10611-t002:** Wave 5 Bayesian MLM informative versus non-informative priors.

Variables	Non-Informative			Informative		
TB	Mean	MC Error	95% Cred. Interval	Mean	MC Error	95% Cred. Interval
Intercept	−2.728	0.004	(−3.409: −2.031)	−2.813	0.023	(−4.275:−1.396)
Marital status: Single	0.277	0.000	(0.134: 0.421)	0.276	0.000	(0.1337: 0.420)
Age: 30–44	1.254	0.000	(1.068: 1.441)	1.254	0.001	(1.068: 1.443)
45–59	1.294	0.000	(1.084: 1.505)	1.296	0.001	(1.085: 1.508)
60+	0.785	0.001	(0.532: 1.037)	0.788	0.001	(0.536: 1.039)
Gender: female	−0.514	0.000	(−0.658: −0.370)	−0.513	0.000	(−0.657: −0.369)
Race: non-black-African	−0.280	0.000	(−0.496: −0.067)	−0.287	0.000	(−0.502: −0.071)
Education: Primary	0.268	0.001	(−0.032: 0.571)	0.269	0.002	(−0.025: 0.569)
Secondary	0.033	0.001	(−0.287: 0.357)	0.038	0.002	(−0.273: 0.357)
Tertiary	−0.248	0.001	(−0.643: 0.146)	−0.242	0.002	(−0.631: 0.148)
Home language literacy: cannot read	0.226	0.001	(−0.072: 0.519)	0.228	0.001	(−0.068: 0.520)
Employment status: not employed	0.032	0.000	(−0.117: 0.182)	0.031	0.001	(−0.118: 0.182)
Regular smoking: No	−0.308	0.000	(−0.470: −0.147)	−0.307	0.001	(−0.467: −0.144)
Other disease: No	−1.056	0.000	(−1.220: −0.891)	−1.056	0.001	(−1.221: −0.889)
Regular exercise; No	0.187	0.000	(0.034: 0.342)	0.186	0.000	(0.033: 0.340)
Consult with health practitioner: No	−0.646	0.000	(−0.849: −0.449)	−0.647	0.000	(−0.849: −0.451)
Diagnosed asthma: No	−0.629	0.001	(−0.878: −0.370)	−0.626	0.002	(−0.875: −0.365)
Diagnosed diabetes: No	0.245	0.001	(−0.040: 0.543)	0.248	0.002	(−0.038: 0.564)
Social grant: No	−0.146	0.000	(−0.296: 0.000)	−0.143	0.000	(−0.293: 0.005)
Perceived household Income: No	0.155	0.000	(0.021: 0.288)	0.152	0.000	(0.020: 0.285)
Access to better housing: No	−0.223	0.000	(−0.376: −0.072)	−0.221	0.000	(−0.374: −0.070)
Income below median: No	0.0318	0.000	(−0.130: 0.1940	0.033	0.000	(−0.128: 0.196)
Expenditure below median: No	0.274	0.000	(0.113: 0.435)	0.272	0.000	(0.111: 0.434)
Geotype: Urban	0.145	0.000	(−0.022: 0.311)	0.142	0.001	(−0.027: 0.311)
Farms	0.027	0.000	(−0.258: 0.301)	0.027	0.001	(−0.257: 0.304)
Province Var(_cons) Tau.prov	6.609	0.014	(1.633: 15.880)	0.587	0.001	(0.231: 1.110)

**Table 3 ijerph-19-10611-t003:** DIC for informative and non-informative.

Prior Type	Dbar	Dhat	pD	DIC
Non-informative	7801.7	7771.82	29.88	7831.58
Informative	7802.73	7769.83	32.894	7835.62

**Table 4 ijerph-19-10611-t004:** Fixed and random effects multilevel analysis of factors associated with TB: two estimation techniques.

Variables	Frequentist MLM		Bayesian MLM (Non-Informative)	
TB	Coef.	95% Conf. Interval	Mean (MC. Error)	95% Cred. Interval
Intercept	0.276 *	(0.133; 0.420)	0.277 (0.000) *	(0.134: 0.421)
Marital status: Single	1.251 *	(1.064; 1.438)	1.254 (0.000) *	(1.068: 1.441)
Age: 30–44	1.293 *	(1.082; 1.504)	1.294 (0.000) *	(1.084: 1.505)
45–59	0.787 *	(0.536; 1.039)	0.785 (0.000) *	(0.532: 1.037)
60+	−0.513 *	(−0.657; −0.369)	−0.514 (0.000) *	(−0.658: −0.370)
Gender:Female	−0.260 *	(−0.475; −0.044)	−0.280 (0.000) *	(−0.496: −0.067)
Race: non-black-African	0.268	(−0.033; 0.569)	0.268 (0.001)	(−0.032: 0.571)
Education: Primary	0.034	(−0.287; 0.353)	0.033 (0.001)	(−0.287: 0.357)
Secondary	−0.245	(−0.638; 0.148)	−0.248 (0.001)	(−0.643: 0.146)
Tertiary	0.227	(−0.068; 0.522)	0.226 (0.001)	(−0.072: 0.519)
Home language literacy: cannot read	0.031	(−0.119; 0.180)	0.032 (0.000)	(−0.117: 0.182)
Employment status: not employed	−0.308 *	(−0.469; −0.147)	−0.308 (0.000) *	(−0.470: −0.147)
Regular smoking: No	−1.051 *	(−1.216; −0.886)	−1.056 (0.000) *	(−1.220: −0.891)
Other disease: No	0.185 *	(0.031; 0.338)	0.187 (0.000) *	(0.034: 0.342)
Regular exercise; No	−0.644 *	(−0.843; −0.445)	−0.646 (0.000) *	(−0.849: −0.449)
Consult with health practitioner: No	−0.633 *	(−0.887; −0.380)	−0.629 (0.001) *	(−0.878: −0.370)
Diagnosed asthma: No	0.238	(−0.055; 0.528)	0.245 (0.001)	(−0.040: 0.543)
Diagnosed diabetes: No	−0.144	(−0.292; 0.004)	−0.146 (0.000)	(−0.296: 0.000)
Social grant: No	0.153 *	(0.019; 0.286)	0.155 (0.000) *	(0.021: 0.288)
Perceived household Income: No	−0.222 *	(−0.374; −0.070)	−0.223 (0.000) *	(−0.376: −0.072)
Access to better housing: No	0.034	(−0.127; 0.196)	0.0318 (0.000)	(−0.130: 0.194)
Income below median: No	0.273 *	(0.113; 0.434)	0.274 (0.000) *	(0.113: 0.435)
Expenditure below median: No	0.147	(−0.021; 0.314)	0.145 (0.000)	(−0.022: 0.311)
Geotype: Urban	0.035	(−0.244; 0.313)	0.027 (0.000)	(−0.258: 0.301)
Farms	0.216 *	(0.079; 0.589)	6.609 (0.014) *	(1.633: 15.880)
LR test	116.17			
Wald Chi2(24)	774.54			
LL	−3902.859			
AIC	7857.78			
BIC	8065.605			

* indicates significant covariates.

## Data Availability

The datasets generated and analyzed during the current study are available upon request from the corresponding author.

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
