# Peer review of "A Multilevel Analysis of the Associated and Determining Factors of TB among Adults in South Africa: Results from National Income Dynamics Surveys 2008 to 2017"

_ijerph, 2022, doi:10.3390/ijerph191710611_

Round 1
Reviewer 1 Report
This study evaluated the predictors of TB. Although the authors explained the detail of statistical method, the method of National Income Dynamics Survey should be also shown briefly.
1. The year of wave 1 to 5 should be shown in Method.
2. LRT, BIC, AIC, and ICC should be spelled out when they appeared first.
3. “Suffering from other diseases” were important factors to predict TB incidence. What are other diseases?
4. Likewise, what does “consultation about health” mean? Does it mean visiting hospitals regularly, reflecting the presence of some diseases? Are there any difference from “Suffering from other diseases”?
5. On line 146, “these age groups were 3.0, 2.5, and 2.5 times more likely to be diagnosed”.
“3.0, 2.5, and 1.5 times” would be correct.
6. Line 154-155, line 196, line 218, line 222, and line 229, I think positive association would be correct because those who had consulted with a health practitioner or those who have asthma were more likely to develop TB. Anyway, these explanations are only confusing for the readers.
7. Line 170-171, how did the authors classify smoking status to regular smokers and non-regular smokers? The association between smoking and TB is not consistent. How did the authors interpret these results?
In addition, smoking status is also usually classified into current smokers, former smokers, and non-smokers, or by using Binkmann index.
8. Line 174-176, “With an OR of 0.336 (coef. -0.09, 95% CI -1.358;-0.822), people who had not consulted with a health practitioner had a 67% lesser chance of being diagnosed with TB than those who had not consulted.” There would be some mistakes.
9. For the readers in other countries, it is difficult to understand “individuals that belonged to households with no access to better housing” on line 199.
10. Line 226-228, Individuals belonging to households with income above the median had higher odds of being diagnosed with TB than people whose household income was below average or median.
This result is inconsistent with the other results about the association between the income and TB. The authors should discuss it in Discussion.
11. Line 251-253, “Not belonging to households with monthly expenditure below the median rendered individuals 1.31 times more likely to be diagnosed with TB than their counterparts.” This result is also inconsistent with the other results about the association between the income and TB. The authors should discuss it in Discussion.
12. In discussion, “The aim was not only to identify TB factors at the individual level but also to determine to what extent the factors vary across such individuals belonging to the same province.”
However, where are the results and discussion on this issue?
13. Likewise, which results show that some form of variation in TB infections is due to the different provinces these individuals belonged on line 314-315?
14. It is difficult to understand the meaning of line 322-324. There are no data on the variability across individuals belonging to the same provinces,
15. On line 324, there are no data on TB treatment.
16. On line 328, there are no data on diabetes.
Author Response
Thank you so much for the feedback. I tried to address all the concerns but lost the 'track changes' at some point. my laptop stopped saving, giving me an error so when I copied on to a new document the changes were not showing on the 'track changes'. If you need to see this document i will gladly share. I have it but it is not the final draft.

Reviewer 2 Report
The study is very interesting and the inclusion of 9-year data records allows an adequate perspective of a public health condition still in force. However, the authors have to attend to some points so that their manuscript has more inference:
a) In the statistical analysis section, it is necessary to indicate for each test applied if the associated assumptions were verified. The information is necessary because the mere fact of applying the technique for hypothesis testing is not enough, it is necessary to verify the assumptions in order to make more robust inferences and ensure reproducibility.
b) Tables 1 and 3 require adjustment of their format since in the revised version the parentheses are on different lines due to the length of the values ​​shown. I also suggest improving, including only one value of standard error and confidence interval. This is simplified by subtracting the lower standard error from the average or subtracting the average from the upper standard error. The same operation to obtain only one confidence interval value. For example, the variable Age: 30-44 has an average value of 1.254, if 1.068 is subtracted from this value, it is equal to 0.186, also if the average is subtracted from the CI 1.441, the value is equal to 0.187. Which indicates that the standard errors or CI are almost symmetric. So instead of using both values, they can just include an error or IC. This will reduce the dimension of the tables.
c) In table 3, I am concerned about the following, there are values ​​of standard error and CI that indicate them as 0.000, this is not correct because there cannot be values ​​of 0. Please review.
d) The discussion is not "contrasted" or despite the redundancy, they are not discussed with other investigations. The lack of related studies is partly understandable, but its background as a theoretical framework is necessary and therefore it is necessary to discuss its contribution and results in the light of other research.
Author Response

(The authors gave the same response as above.)

Round 2
Reviewer 1 Report
The manuscript was revised enough.
Reviewer 2 Report
The authors made the adjustments indicated in the previous revision to the manuscript.
This manuscript is a resubmission of an earlier submission. The following is a list of the peer review reports and author responses from that submission.